# Psychosocial Care for Adult Cancer Patients: Guidelines of the Italian Medical Oncology Association

**DOI:** 10.3390/cancers13194878

**Published:** 2021-09-29

**Authors:** Caterina Caminiti, Francesca Diodati, Maria Antonietta Annunziata, Paola Di Giulio, Luciano Isa, Paola Mosconi, Maria Giulia Nanni, Adele Patrini, Michela Piredda, Claudia Santangelo, Claudio Verusio, Michela Cinquini, Veronica Andrea Fittipaldo, Rodolfo Passalacqua

**Affiliations:** 1Clinical and Epidemiology Research Unit, University Hospital of Parma, 43126 Parma, Italy; ccaminiti@ao.pr.it (C.C.); fdiodati@ao.pr.it (F.D.); 2Oncological Psychology Unit, Centro di Riferimento Oncologico di Aviano (CRO) IRCCS, 33081 Aviano, Italy; annunziata@cro.it; 3Department of Public Health and Pediatrics, University of Turin, 10124 Turin, Italy; paola.digiulio@unito.it; 4Division of Oncology, Hospital of Melegnano, 20064 Gorgonzola, Italy; luciano.isa@tiscali.it; 5Laboratory for Medical Research and Consumer Involvement, Department of Public Health, Istituto di Ricerche Farmacologiche Mario Negri IRCCS, 20156 Milan, Italy; Paola.Mosconi@marionegri.it; 6Department of Neuroscience and Rehabilitation, Institute of Psychiatry, University of Ferrara, 44121 Ferrara, Italy; nnnmgl@unife.it; 7Associazione C.A.O.S. Varese, FAVO Comitato Regionale Lombardia, 21041 Albizzate, Italy; caosvarese@gmail.com; 8Research Unit of Nursing Science, Campus Bio-Medico University of Rome, 00128 Rome, Italy; m.piredda@unicampus.it; 9Associazione Vivere Senza Stomaco Si Può Onlus, 44123 Ferrara, Italy; claudia.santangelo@viveresenzastomaco.org; 10Department of Medical Oncology, Presidio Ospedaliero di Saronno, ASST Valle Olona, 21047 Saronno, Italy; claudioverusio@gmail.com; 11Istituto di Ricerche Farmacologiche Mario Negri IRCCS, 20156 Milan, Italy; michela.cinquini@marionegri.it (M.C.); veronicaandrea.fittipaldo@marionegri.it (V.A.F.); 12Medical Oncology Division, Department of Oncology, ASST of Cremona, 26100 Cremona, Italy

**Keywords:** psychosocial care, psychosocial needs, supportive care, quality of life, guidelines, cancer

## Abstract

**Simple Summary:**

People with cancer often experience psychological and social needs, related to their disease and treatments, that can negatively impact quality of life. Various social interventions can be helpful but are not always offered to patients who would benefit from them. This guideline aims to help oncology professionals address the psychosocial aspects of their adult patients and of those who care for them. It was compiled by a multidisciplinary panel, including patients, using rigorous methodology. Topics include patient information and communication, screening and management of psychosocial needs, and caregiver support. In particular, evidence emphasizes that nurses play a central role in providing psychosocial care and information for cancer patients, and that Physician communication skills must be improved with specific evidence-based training. In addition, psychosocial needs must be promptly detected and managed, especially with appropriate non-pharmacological interventions.

**Abstract:**

Psychosocial morbidity can have negative consequences for cancer patients, including maladaptive coping, poor treatment adherence, and lower quality of life. Evidence shows that psychosocial interventions can positively impact quality of life, as well as symptoms and side effects; however, they are not always offered to patients who might benefit from them. These guidelines were produced by a multidisciplinary panel of 16 experts, including patients, following GRADE methodology. The panel framed clinical questions and voted on outcomes to investigate. Studies identified by rigorous search strategies were assessed to rate certainty of evidence, and recommendations were formulated by the panel. Although the quality of the evidence found was generally moderate, interventions could be recommended aimed at improving patient information, communication with healthcare professionals and involvement in decision-making; detecting and managing patient psychosocial needs, particularly with non-pharmacological therapy; and supporting families of patients with advanced cancer. The role of nurses as providers of information and psychosocial care is stressed. Most recommended interventions do not appear to necessitate new services or infrastructures, and therefore do not require allocation of additional resources, but predominantly involve changes in clinical staff behavior and/or ward organization. Patients should be made aware of psychosocial care standards so that they can expect to receive them.

## 1. Introduction

The significant impact of cancer on the lives of patients and their families is not restricted to symptoms and treatment side effects, but may extend to a wide range of psychological, emotional, social, cultural and spiritual aspects of health [1,2,3]. For example, the prevalence of spectrum depressive disorder in cancer patients has been reported to be up to 2–3 times higher than in the general population [3]. The burden of cancer is also aggravated by the frequent practical difficulties that patients encounter during the course of treatment and by the lack of adequate information on their state of health [1,4,5]. Nevertheless psychosocial needs are often not recognized by the clinical team and thus remain unmet [4,5,6]. This has important implications for both patients and their families, as shown by numerous studies reporting the association of psychosocial morbidity with maladaptive coping, reduced quality of life (QoL), impaired social relationships, suicide risk, longer rehabilitation times, poor treatment adherence and abnormal pathological behavior, familial dysfunction and possibly shorter survival [3]. Therefore, cancer care cannot neglect psychosocial care [7,8], which is defined as any activity aimed at improving or reducing the impact of cancer on mental health and improving patients’ abilities to cope with the demands of treatment and uncertainty of disease outcome across the entire spectrum from pre-diagnosis to palliative care and survival [9]. In this regard, a patient-centered, multidisciplinary approach to cancer provided by a team of healthcare professionals from different fields could improve patients’ quality of life [10,11]. In particular, several studies have shown the positive health impact of a wide range of psychosocial interventions [12,13,14,15], but many patients who could benefit from such interventions still do not receive them [7,9,16] due to financial and organizational barriers. Such barriers may be exacerbated by healthcare professionals’ reduced knowledge and understanding of the key role that psychosocial care plays in supporting the biomedical treatment of cancer patients [17]. For this reason, it is essential to encourage research on strategies for the implementation of psychosocial interventions and the creation of validated guidelines to promote the adoption and integration of psychosocial care in routine practice [7,18].

In this challenging scenario, the Italian Association of Medical Oncology (AIOM) has developed these guidelines, following a rigorous methodology, with the aim of ensuring a change in the behavior of professionals working in cancer departments. Overall objectives are to: ensure detection and management of psychosocial needs in people with cancer; reduce barriers to the provision of care to patients of different linguistic and cultural backgrounds; highlight gaps in the research evidence. After assessment of compliance with methodological quality standards by the Ministry of Health, the guidelines have been included in the National Guideline System [19], and therefore should be used by healthcare professionals in Italy according to current law.

## 2. Materials and Methods

Following AIOM indications, these guidelines were produced by a multidisciplinary and multiprofessional panel comprising 16 experts, including oncologists, epidemiologists, psychiatrists, psychologists, nurses and patient representatives. The panel was supported by the Evidence Review Group of the Mario Negri Istituto di Ricerche Farmacologiche IRCCS of Milan, which facilitated methodology compliance, trained panelists, assessed evidence and guided the process from evidence search to recommendation formulation. Since 2016, AIOM guidelines have included recommendations obtained with the GRADE approach [20]. Relevant clinical questions were framed according to PICO (Population, Intervention, Control, and Outcomes). Importance of possible outcomes was rated by each panelist numerically on a 1–9 scale (7–9, critical; 4–6, important; and 1–3, of limited importance); outcomes with an average rating from 7 to 10 were retained. A comprehensive, reproducible bibliographic search for each clinical question was run on the MedLine and Embase databases (see Appendix A for the search strategies), and title and abstract were screened by the panel to select potentially relevant literature. According to GRADE, the certainty of evidence was then rated on the following 5 main dimensions applied to the individual outcome: risk of bias (limitations in study design), inconsistency (heterogeneity among study results), indirectness (direct applicability of results between PICO—population, intervention, comparison, outcomes—and retrieved evidence), imprecision (width of confidence interval around the point estimate, related to optimal information size) and publication bias (probability that evidence was published depending on the nature and direction of results). Certainty of evidence was then synthesized into four levels (very low, low, moderate, high), as shown in Table 1. The panel then decided on the strength of each recommendation, which reflects not only the certainty but also the clinical relevance of evidence. In accordance with AIOM indications, strength of recommendation has been worded as reported in Table 2. The final document was reviewed by two experts (a methodologist and an oncologist) appointed by AIOM who were external to the development process. 

## 3. Questions and Recommendations

This paragraph reports the clinical questions, a summary of the evidence, and corresponding recommendations. Table 3 displays all recommendations with quality of the evidence and strength. An attempt was made to order clinical questions following the patient’s diagnostic–therapeutic trajectory.

### 3.1. Question 1: In Patients with Cancer, Is Information Support Provided by Ward Nursing Staff Indicated?

The literature highlights the importance of the nurse as provider of psychosocial care. A Cochrane systematic review [21] assessed the effects of psychosocial interventions, involving interpersonal dialogue between a “trained helper” and individual newly diagnosed cancer patients, to improve QoL and general psychological distress in the 12 months following an initial cancer diagnosis. Thirty RCTs including 5155 patients were considered. Meta-analysis on 9 trials and 1249 patients detected very small overall effects on QoL; however, in the subgroup analysis by discipline of trained helper, only interventions carried out by nurses showed significantly positive results (Standardized Mean Difference (SMD) = 0.23; 95% CI 0.04 to 0.43). The evidence on the role of nurses was derived from RCTs conducted in the United States on patients with breast cancer, which may reduce generalizability. Another Cochrane review [22] assessed the effectiveness on QoL of breast-cancer patients from interventions provided by specialist nurses, professionals who accompany patients through their care path and give information according to individual needs. Given the heterogeneity of reported interventions, a narrative synthesis is presented. Positive effects were observed on some dimensions of QoL, such as anxiety and early recognition of depressive symptoms. Specifically, in an RCT on 172 patients 20 months after mastectomy, in the intervention arm 92% of women were free from anxiety and 95% from depression, vs. 70% for both outcomes in the control arm.

Recommendation: In patients with cancer, information support provided by ward nursing staff should be considered throughout the care process. 

### 3.2. Question 2: Is It Advisable to Implement a Service in the Ward to Ensure That All Cancer Patients Who So Wish Can Receive the Information and Related Support They Need?

An Italian RCT [23] involving 38 cancer centers assessed the effectiveness of a Point of Information and Support (PIS), a library for patients and their families containing selected information material, managed by trained nursing staff. Primary endpoint was psychological distress, measured during index days on patients consecutively accessing the centers, regardless of PIS use. A total of 3197 patients were included. Intention-To-Treat analysis did not detect any effect; however, 52% of centers in the PIS arm did not implement the intervention according to protocol. In fact, analysis restricted to centers that enacted the intervention showed a 20% reduction of the rate of patients with distress (OR = 0.80; 95% CI 0.62 to 1.03; *p* = 0.09). However, this finding is not statistically significant due to low statistical power. Failure to consider intervention complexity and the effect of intracluster variation at the design stage are main limitations of this RCT.

Recommendation: In patients with cancer, a point of Information and Support in the ward could be considered.

### 3.3. Question 3: Does Providing a List of Possible Questions to Patients Improve Patient–Physician Communication?

Question Prompt Lists (QPL) are lists of possible questions that the patient may ask his/her physician, selected from the analysis of medical consultations or by focus groups. A systematic review [24] examined the effectiveness of QPL interventions on communication, psychological and/or cognitive outcomes of cancer patients. Sixteen studies were included, of which there were 15 RCTs on 2330 patients. Cancer type was diverse across studies, and patients were predominantly seeing their provider regarding treatment for the first time. The number of questions asked during consultation was significantly higher in the intervention group in 4/6 trials. In addition, 2/9 trials found that the QPL reduced anxiety a few weeks after use, and 2 found anxiety significantly increased immediately after the consultation, to decrease again at 1 week. Finally, of the 4 studies investigating effects on recall, 2 reported a statistically significant effect when the physician endorsed QPL use. Heterogeneity between interventions precluded measure of overall effect. 

Recommendation: A list of possible questions should be given to patients with cancer during their first visits and its use should be encouraged by the oncologist.

### 3.4. Question 4: Is the Use of Tools to Favor Patient Involvement in Decision-Making at Crucial Time Points of Care Indicated?

Decision aids (DAs) are evidence-based tools that support patients’ choices about their care, by making their decisions explicit and helping them clarify congruence between decisions and personal values. A meta-analysis of 34 RCTs concerning the use of different types of DAs in oncology [25] reported significant improvement of knowledge about available options, both in screening (weighted average effect size (ES) = 0.50; 95% CI 0.27 to 0.73; *p* < 0.0001), and prevention/treatment (weighted average ES = 0.50; 95% CI 0.31 to 0.70; *p* < 0.0001) contexts. A limitation reported for most included studies was the lack of details on subject allocation to intervention and control arms. 

Positive effects of DAs are confirmed by a Cochrane systematic review [26], which considered 105 trials involving 31,043 patients on the use of DAs in screening and treatment contexts for various diseases, predominantly cancer. This review also detected increase in knowledge (Mean Difference (MD) 13.27/100; 95% CI 11.32 to 15.23; 52 trials; *n* = 13,316 patients; high-quality evidence), accuracy of risk perception (RR 2.10; 95% CI 1.66 to 2.66; 17 trials; *n* = 5096 patients; moderate-quality evidence), and congruence between values and care decisions (RR 2.06; 95% CI 1.46 to 2.91; 10 trials; *n* = 4626 patients; low-quality evidence), for DA use vs. usual care. Important limitations highlighted by the authors concern the wide variability between included studies in terms of contexts, intervention types and outcomes.

Recommendation: Tools aimed at fostering patient involvement in the decision-making process (Decision Aids) could be considered during crucial phases of care. 

### 3.5. Question 5: Can Communication Training Addressed to Patients Favor Participation and Communication during the Visit?

A systematic review [27] examined the effects of Patient Training Programs on communication in different contexts. Thirty-two trials were considered (19 RCTs and 13 quasi-experimental), 9 of which were in oncology. Eight out of ten trials measuring patient participation during visits detected a significant difference between intervention and control groups, or before and after the intervention. Notably, 4/5 studies revealed significant differences in the ability to express concern. Regarding communication process outcomes, 7/10 trials found that trained patients had received significantly more information during consultations. Five/six trials that considered health or treatment outcomes obtained non-significant or mixed results. In addition, 5/6 trials that considered psychosocial outcomes did not detect a difference or reported mixed results. Of the 32 included studies, 28 (88%) were conducted in the U.S., which may limit generalizability. The review did not report overall data but only frequencies detected in individual studies.

Recommendation: Training interventions aimed at patients could be offered to encourage their active participation in communication with health professionals.

### 3.6. Question 6: Is Communication Skills Training for Healthcare Professionals Effective in Improving Healthcare Professional Communication Outcomes? 

A Cochrane review [28] assessed whether communication skills training (CST) is effective in changing behavior of healthcare professionals (HCPs) working in cancer care. The third update of the review included 17 RCTs involving 1240 HCPs and the analysis of 2648 consultations. Trained HCPs were more likely to use open questions in the post-intervention interviews (5 studies, 796 participant interviews; SMD 0.25, 95% CI 0.02 to 0.48; *p* = 0.03; and to show empathy (6 studies, 844 participant interviews; SMD 0.18, 95% CI 0.05 to 0.32; *p* = 0.008, and less likely to give facts only without individualizing their responses to the patient’s emotions or offering support (5 studies, 780 participant interviews, SMD 0.26, 95% CI 0.51 to 0.01; *p* = 0.05). It was not possible to determine whether the effect persisted with time, whether consolidation sessions would be needed, or which type of intervention worked best.

Recommendation: All oncologists should attend a structured course according to available scientific evidence, of at least 3 days, to improve communication skills.

### 3.7. Question 7: Is the Use of a Screening Intervention for Distress Indicated in Patients with Cancer?

A Cochrane review [29] examined the effectiveness and safety of screening of psychosocial well-being and care needs of people with cancer. Eligible studies were RCTs or non-randomized controlled trials on adult cancer patients, considering Outcomes collected with validated self-report questionnaires or through interviews with the use of validated Patient Reported Outcome Measures (PROMs). Twenty-six studies (18 RCTs) were included, on a total of 7654 subjects. Meta-analysis on 3 studies (2 RCTs) revealed no beneficial effect of screening interventions on Health-Related Quality of Life (HRQoL) (MD 1.65, 95% CI −4.83 to 8.12, 2 RCTs, 6 months follow-up); distress (MD 0.0, 95% CI −0.36 to 0.36, 1 RCT, 3 month follow-up); care needs (MD 2.32, 95% CI −7.49 to 12.14, 2 RCTs, 3 month follow-up). The authors stressed the overall high risk of bias of most included studies.

Recommendation: In patients with cancer, screening for psychological distress should be considered, and different management programs should be implemented according to the level of distress.

### 3.8. Question 8: In Patients Exhibiting Distress Resulting from and/or Concomitant with Their Active Cancer Illness, Is the Use of Non-Pharmacological Therapy, i.e., Based on Psychosocial and Psychological Interventions, Indicated? 

A large systematic review and meta-analysis of 71 RCTs (13,098 patients) [30] estimated the overall ES of psychological interventions on anxiety and distress in cancer patients. Fifty-one studies were included in the meta-analysis. Overall, 87% (61/71) of trials reported a reduction of anxiety scores from baseline. Overall ES was −0.21 (95% CI −0.30 to −0.13). Subgroup analysis stratified by intervention characteristics showed higher ESs for studies that provided relaxation training (*n* = 12; ES: −0.53), individual delivery (*n* = 24; ES: −0.32), face-to-face with self-delivery mode (*n* = 7; ES: −0.35), or deliver by multiple providers (psychologist and psychiatrist, *n* = 2; ES: −0.40). Only studies reporting anxiety outcomes separately from distress were included, which might have led to publication bias. In addition, ES was only estimated using immediate post-intervention outcomes, and therefore it could not be determined whether benefit was sustained.

An individual patient data meta-analysis [31] evaluated the effects of psychosocial interventions on QoL, on the emotional function (EF), and social function (SF) of patients with cancer. A total of 61 eligible RCTs were identified, and their Principal Investigators (PIs) were invited to share anonymized patient data. Data were obtained for 22 trials, including 4217 patients, 2215 in the intervention and 2002 in the control group. Psychosocial interventions significantly improved QoL (β = 0.14, 95% CI 0.06 to 0.21, *p* = 0.05), EF (β = 0.13, 95% CI 0.05 to 0.20, *p* < 0.01), and SF (β = 0.10, 95% CI 0.03 to 0.18, *p* = 0.05). Effects were greater for younger subjects and in studies targeting patients with distress. Based on these findings, the authors underlined the importance of offering interventions to patients who are most likely to benefit, also in consideration of limited available resources. Among limitations, authors noted the large heterogeneity between included trials, and that only 36% of RCTs could be included in the meta-analysis.

Recommendation: Non-pharmacological interventions aimed at reducing cancer-related distress should be targeted and offered to patients who are most likely to benefit.

### 3.9. Question 9: In Patients Exhibiting Depressive Disorders Resulting from and/or Concomitant with Their Active Cancer Illness, Is the Use of Non-Pharmacological Therapy, i.e., Based on Psychosocial and Psychological Interventions, Indicated?

A systematic review and meta-analysis [32] assessed the effects of pharmacological or psychological interventions aimed at cancer patients with major depression or depressive symptoms. Nine of the twenty-five included RCTs concerned psychological interventions. In these, a psychological intervention was compared with a pharmacological intervention, with another psychological intervention or with usual care. Primary outcome was the mean difference of scores on a validated depression rating scale post-treatment with respect to baseline. Despite the high level of heterogeneity of included studies, results were statistically significant in favor of the experimental group (SMD 1.40, 95% CI −2.50 to −0.29, *p* = 0.01, I2 = 96%). However, the significant difference did not persist at 6–8 months’ follow-up in 4 of 6 trials. Studies on psychological interventions did not report data on harm.

A systematic review and meta-analysis [33] evaluated whether non-pharmacological interventions reduced depressive symptoms among breast cancer patients. RCTs with a non-intervened control group on non-terminal patients with no current psychiatric illness were eligible. Forty-one RCTs on 4869 patients were included. Non-pharmacological interventions significantly reduced depressive symptoms (SMD −0.516, 95% CI −0.814 to −0.218). The effect was statistically significant for psychotherapy (SMD −0.819, 95% CI −1.608 to −0.030, *p* = 0.042) and yoga (SMD −0.385, 95% CI −0.633 to −0.136, *p* = 0.002), when the heterogeneity was reduced. It should be noted that time since diagnosis was not considered, and that no follow-up data were available. 

Recommendation: In cancer patients, non-pharmacological interventions aimed at reducing depressive disorders related to cancer should be considered for patients who are most likely to benefit.

### 3.10. Question 10: In Patients Exhibiting Depressive Disorders Resulting from and/or Concomitant with Their Active Cancer Illness, Is the Use of Psychopharmacological Therapy Indicated?

A Cochrane review [34] assessed the efficacy, tolerability and acceptability of antidepressants for treating depressive symptoms in adults with cancer of any site and stage. Ten RCTs (885 patients) were included, 7 of which contributed to the meta-analysis for the primary outcome, depressive symptom reduction. For acute-phase treatment response (6–12 weeks), no difference was found between antidepressants and placebo on symptoms of depression, measured both as a continuous outcome (SMD 0.13; 95% CI 1.01 to 0.11, 5 RCTs, 266 participants; very-low-certainty evidence), and as a proportion of people who had depression at the end of the study (risk ratio RR=0.82; 95% CI 0.62 to 1.08, 5 RCTs, 417 participants; very-low-certainty evidence). As for tolerability, no statistically significant differences were observed between antidepressants and placebo in terms of dropout for side effects, RR 1.19 (95% CI 0.54-2.62, 7 RCTs, 479 participants). Among limitations, it should be noted that included studies did not report follow-up data beyond 12 weeks, and risk of bias was rated unclear or high. 

Recommendation: In cancer patients exhibiting depressive disorders, pharmacological interventions could be considered as a therapeutic option.

### 3.11. Question 11: Should Systematic Screening for Patient Psychosocial Needs Be Performed in Cancer Wards, with Activation of a Structured Response Strategy? 

The Cancer Journey Action Group of the Canadian Partnership Against Cancer (CPAC) and the Association of Psychosocial Oncology (CAPO) conducted a systematic review [1] to develop recommendations on routine assessment of psychosocial and supportive care needs. A total of nine clinical practice guidelines, three systematic reviews, and 14 primary studies (five RCTs) were included. Overall, findings seem to suggest an impact on various outcomes, especially pertaining to communication and discussion on aspects of QoL. Studies differed by design, outcomes and screening approaches.

A systematic review [35] including 9 trials (7 RCTs and 2 quasi-randomized) examined the effectiveness of interventions designed to reduce unmet needs among cancer patients. Interventions were delivered face-to-face, by telephone, or as a combination, by different professionals according to the study. Six of the nine studies (4 large high-quality RCTs) reported no intervention effect, while 3 reported a small effect. 

Recommendation: In patients with cancer, screening for psychosocial needs could be considered in clinical practice.

### 3.12. Question 12: For Cancer Patients of Different Linguistic and Cultural Backgrounds, Can the Presence of Cultural Brokers and/or Interpreters Improve Patient-Healthcare Professional Communication? 

A systematic review [36] evaluated the use of language services for patients with limited English proficiency receiving palliative care. Ten studies (six qualitative, four quantitative) were included. All found that the quality of care was influenced by the type of interpreter. When professional interpreters were not used, patients and families had inadequate understanding about diagnosis and prognosis during goals of care conversations, and patients had worse symptom management. Furthermore, the 6 studies where providers relied on family to interpret important information about diagnosis and prognosis concluded that this practice led to poor communication and negative outcomes. In particular, one study reported the negative effects of involving children as interpreters, including burnout, maladaptive behavior, and truancy within the families. Half of the studies concluded that professional interpreters were not adequately utilized, suggesting that pre-meetings between clinicians and interpreters are important to discuss topics and terminology to be used during visits. Limitations of this review include the lack of quantitative analysis and statistical correlation calculation, due to study heterogeneity. In addition, no RCTs were available, and most studies were single-center and with small sample sizes.

Recommendation: To improve the care of patients of different languages and cultures, the presence of cultural mediators or interpreters with a "cultural competence" should be activated.

### 3.13. Question 13: Can Cultural Competence Education for Healthcare Professionals Improve Communication with Cancer Patients of Different Linguistic and Cultural Backgrounds?

A Cochrane review [37] assessed the effects of cultural competence education intervention for HCPs, considering effects on patients with different pathologies, including cancer. Five RCTs were included, involving 337 HCPs, and 8400 patients of whom at least 3463 (41%) had culturally and linguistically diverse backgrounds. A low-quality trial conducted in the U.S. reported that the intervention significantly improved health behavior (client concordance with attendance) compared with controls (relative risk RR = 1.53, 95% CI 1.03–2.27). A low-quality trial from The Netherlands observed an improvement of involvement in care by "non-Western" patients with Western doctors, in terms of mutual understanding (SMD = 0.21, 95% CI 0.00–0.42. No differences were found for treatment outcomes. Study heterogeneity precluded meta-analyses.

Recommendation: Healthcare professionals could participate in cultural competence education programs in order to foster communication with patients of different languages and cultures. 

### 3.14. Question 14: For Families of Advanced Cancer Patients, Are Supportive Psychosocial Interventions Indicated? 

A Cochrane review [38] including 11 RCTs on 1836 informal caregivers of patients with terminal illness assessed the effect of supportive interventions, delivered directly to caregivers or indirectly via patient care. There was low-quality evidence that direct interventions significantly reduce psychological distress in the short term (8 studies: SMD = −0.15; 95% CI −0.28 to −0.02), while no statistically significant effect was found on coping skills and QoL. The two trials on indirect interventions assessed different outcomes and thus no overall estimation was possible; however, authors concluded that they might have protective effects on distress. Limitations included poor reporting of methods in considered studies and absence of data on possible adverse events.

An RCT on 160 caregivers of patients with life expectancy < 6 months [39] tested the effectiveness of existential behavioral therapy vs. usual care on mental stress and quality of life, also in the long-term (at 12 months). Multivariate analysis showed a statistically significant, clinically moderate favorable effect on depression (*p* = 0.04) and QOL, especially at 12 months (*p* = 0.002). The 12-month effects were clinically relevant, as the distress values improved from abnormal to within the normal range. A possible limitation noted by the author was the heterogeneity of the sample. As almost half of the patients died during their stay at the palliative care unit (average 10 days), the majority of caregivers were already grieving at the beginning of the intervention.

Recommendation: Family members of patients diagnosed with advanced cancer should be able to receive supportive interventions.

## 4. Discussion

This paper describes the current version of the first Italian guidelines on psychosocial assistance to adult cancer patients, which have been updated annually by the Italian Medical Oncology Association since 2012. To our knowledge, no similar comprehensive guidelines produced following a rigorous validation processes have been published in the past decade.

A strength of this document is that it provides practical guidance to assist oncology professionals in treating the psychosocial well-being of all patients throughout the course of their illness. Most peer-reviewed publications in this field focus on specific psychosocial problems (e.g., anxiety or depression), or concern selected populations by stage of disease (e.g., rehabilitation, end of life) or by type of cancer. Furthermore, these guidelines were developed following a rigorous methodology and taking into account the different perspectives of the multidisciplinary and multiprofessional panel, including patient representatives. The contribution of the latter was particularly relevant, as they were recognized by the panel from the design stage as fully qualified experts to identify the problems that have priority for patients, inform if the effects are significant, assess risks and benefits, and evaluate acceptability and feasibility [40,41]. 

Our work has some limitations. Firstly, although recommendations were based on the most rigorous evidence available, it should be noted that it is not always possible to produce evidence of efficacy in the psychosocial field with the standard methodologies used for pharmaceutical clinical trials. Indeed, conducting high-quality RCTs can be difficult due to the lack of standardized interventions that hinder comparability, ethical concerns when using controls, challenges with blinding, and frequent funding and recruitment issues [21,42,43]. In light of these challenges, the lack of strong evidence in psychosocial research does not necessarily correspond to an absence of recommendation. Secondly, evidence search was restricted to the English language, and the grey literature was not considered, which may have caused us to miss relevant studies. Finally, recommendations were formulated with the Italian context in mind, which could limit generalizability; however, they are supported by evidence from studies conducted all over the world, which makes them applicable to other countries as well.

In general, clinical practice guidelines are aimed at fostering the integration of interventions into routine care, as they present critically evaluated evidence in a way that allows research evidence to be translated into practice. These guidelines in particular can be crucial for informing and educating those who do not have psychosocial expertise, potentially increasing the status of psycho-oncology [44].

Although questions were investigated separately, the authors stress the importance of multimodal and multidisciplinary approaches to address psychosocial issues of individuals with cancer, as recommended interventions are closely connected and act synergistically on patient health. 

Regarding resource implications, it is worth noting that many of the recommended interventions do not require new services or infrastructures, and therefore do not involve the allocation of additional resources, but mainly changes in clinical staff behavior. Therefore, implementation in practice may only require organizational and cultural changes.

At the time of writing, the COVID-19 outbreak is still ongoing. The heavy impact of this unprecedented health crisis on people with cancer has been described by various authors [45,46,47,48]. Difficulty in accessing treatment, cancer recurrence and progression due to delay in treatment have been the most important concerns among cancer patients. These concerns can produce additional mental health distress and significant psychological effects. When this guideline was finalized in late 2019, however, the outbreak had not yet begun, so it did not investigate approaches and strategies for dealing with psychological distress during emergencies such as the one currently underway. For the next guideline update, the panel has identified a new question concerning the psychosocial impact of care interventions provided at a distance, such as telemedicine and remote consultation. These modalities emerged as a solution to ensure continuity of care for cancer patients while avoiding unnecessary hospital access and thus lowering the risk of infection. If proven effective and well-accepted by patients, however, they could become alternatives offered in routine practice, also outside of emergencies.

The panel recognizes that other relevant questions would deserve consideration, among others the effects of lifestyle interventions (such as dietary education and physical activity) on the quality of life of cancer patients as well as their psychological and social conditions. These topics may be gradually included in future versions of the guidelines.

## 5. Conclusions

Guideline dissemination to healthcare professionals is a first step towards promotion of psychosocial care awareness and implementation. It is also essential that patients and their families know that standards for psychosocial care are available, and expect their implementation at the hospital where they are cared for. For this reason, the panel is currently working on a patient version of the guidelines, in collaboration with patient associations. The resulting document is intended to be the translation, in an accessible language, of the recommendations developed for clinicians.

## Figures and Tables

**Table 1 cancers-13-04878-t001:** Grading of certainty of evidence.

Certainty of Evidence	Meaning	Consequence
High	High confidence in results	It is very likely that the true effect of the treatment is similar to the estimated one
Moderate	Moderate confidence in results	It is likely that the true effect of the treatment is similar to the estimated one but there is still the possibility that the effect is different
Low	Results are not trustworthy	Confidence in the effect estimate is limited: the true effect could be substantially different from the estimated one
Very low	Results are totally not trustworthy	Confidence in the effect estimate is very limited: it is likely that the true effect is substantially different from the estimated one

**Table 2 cancers-13-04878-t002:** Strength of recommendation according to the GRADE adaptation for AIOM.

Strength of Recommendation	Meaning
Strong for	The intervention should be considered as the first treatment option (benefits are higher than risks)
Conditional for	The intervention can be considered as a possible treatment option (not sure that benefits are higher than risks)
Conditional against	The intervention should not be considered as the first treatment option; it could be considered in selected cases after discussion with the patient (not sure that risks are higher than benefits)
Strong against	The intervention must not be considered as a possible treatment option (risks are higher than benefits)

**Table 3 cancers-13-04878-t003:** Clinical recommendations for integrated psychosocial interventions into the routine care of patients with cancer.

Question	Certainty of the Evidence	Recommendation	Strength of the Recommendation
Question 1: In patients with cancer, is information support provided by ward nursing staff indicated?	**Moderate**	In patients with cancer, information support provided by ward nursing staff should be considered throughout the care process [21,22]	**Strong for**
Question 2: Is it advisable to implement a service in the ward to ensure that all cancer patients who so wish can receive the information and related support they need?	**Moderate**	In patients with cancer, a Point of Information and Support in the ward could be considered [23]	**Conditional for**
Question 3: Does providing a list of possible questions to patients improve patient-physician communication?	**Moderate**	A list of possible questions should be given to patients with cancer during their first visits and its use encouraged by the oncologist [24]	**Strong for**
Question 4: Is the use of tools to favor patient involvement in decision-making at crucial time points of care indicated?	**Moderate**	Tools aimed at fostering patient involvement in the decision-making process (Decision Aids) could be considered during crucial phases of care [25,26]	**Conditional for**
Question 5: Can communication training addressed to patients favor participation and communication during the visit?	**Moderate**	Training interventions aimed at patients could be offered to encourage their active participation in communication with health professionals [27]	**Conditional for**
Question 6: Is communication skills training for healthcare professionals effective in improving healthcare professional communication outcomes?	**Moderate**	All oncologists should attend a structured course according to available scientific evidence, of at least 3 days, to improve communication skills [28]	**Strong for**
Question 7: Is the use of a screening intervention for distress indicated in patients with cancer?	**Low**	In patients with cancer, screening for psychological distress should be considered and different management programs should be implemented according to the level of distress [29]	**Strong for**
Question 8: In patients exhibiting distress resulting from and/or concomitant with their active cancer illness, is the use of non-pharmacological therapy, i.e. based on psychosocial and psychological interventions, indicated?	**Moderate**	Non-pharmacological interventions aimed at reducing cancer-related distress should be targeted and offered to patients who are most likely to benefit [30,31]	**Strong for**
Question 9: In patients exhibiting depressive disorders resulting from and/or concomitant with their active cancer illness, is the use of non-pharmacological therapy, i.e. based on psychosocial and psychological interventions, indicated?	**Moderate**	In cancer patients, non-pharmacological interventions aimed at reducing depressive disorders related to cancer should be considered for patients who are most likely to benefit [32,33]	**Strong for**
Question 10: In patients exhibiting depressive disorders resulting from and/or concomitant with their active cancer illness, is the use of psychopharmacological therapy indicated?	**Low**	In cancer patients exhibiting depressive disorders, pharmacological interventions could be considered as a therapeutic option [34]	**Conditional for**
Question 11: Should systematic screening for patient psychosocial needs be performed in cancer wards, with activation of a structured response strategy?	**Moderate**	In patients with cancer, screening for psychosocial needs could be considered in clinical practice [1,35]	**Conditional for**
Question 12: For cancer patients of different linguistic and cultural backgrounds, can the presence of cultural brokers and/or interpreters improve patient-healthcare professional communication?	**Low**	To improve the care of patients of different languages and cultures, the presence of cultural mediators or interpreters with a "cultural competence" should be activated [36]	**Strong for**
Question 13: Can cultural competence education for healthcare professionals improve communication with cancer patients of different linguistic and cultural backgrounds?	**Low**	Healthcare professionals could participate in cultural competence education programs in order to foster communication with patients of different languages and cultures [37]	**Conditional for**
Question 14: For families of advance cancer patients, are supportive psychosocial interventions indicated?	**High**	Family members of patients diagnosed with advanced cancer should be able to receive supportive interventions [38,39]	**Strong for**

## Data Availability

The data presented in this study are available on request from the corresponding author.

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
