# Peer review of "Psychosocial Care for Adult Cancer Patients: Guidelines of the Italian Medical Oncology Association"

_cancers, 2021, doi:10.3390/cancers13194878_

Round 1

Reviewer 1 Report

Caminiti and colleagues proposed a research article presenting the guidelines of the         
Italian Medical Oncology Association describing the approaches adopted for the psychosocial care of cancer patients. The authors clearly described the clinical questions posed by a multidisciplinary team composed of 16 experts, including clinicians and healthcare professionals providing also further recommendations to improve the psychosocial management of cancer patients. Overall, the manuscript is interesting and well-written, however, there is some missing information that the authors have to address before publication:
1) In the Introduction section, the authors should introduce how a multidisciplinary approach to cancer treatment provided by both clinicians and health care professionals and the adoption of healthy lifestyles could ameliorate patients’ quality of life. For this purpose, please see:
- PMID: 34132354
- PMID: 33061625
- PMID: 33396551
- PMID: 32266126
- PMID: 33437520
2) Please add commas for thousands (e.g. 5155 should be 5,155; etc.);
3) The Discussion section should be significantly improved. The authors should add information about the impact of the ongoing COVID-19 pandemic for cancer patients highlighting the effects of social distancing and hospital restrictions on the psychosocial sphere of patients. It is important to describe the approach adopted for the treatment of patients during the pandemic and which strategies were implemented to avoid psychological distress for both patients and caregivers. For this purpose, please see:
- PMID: 33950368
- PMID: 32785162
- PMID: 33430131
- PMID: 33897477
4) In the Discussion section, please briefly discuss how dietary interventions and physical activity could improve the quality of life of cancer patients as well as their psychological and social conditions. Do the Italian Medical Oncology Association guidelines consider this type of interventions in support of the psychosocial sphere of cancer patients? Please clarify this point.

Author Response

We sincerely thank the reviewer for his/her time and for the relevant comments and suggestions which have enabled us to improve our paper. Please see the attachment for a point-by-point response. 

Reviewer 2 Report

Thank you for the opportunity to review the manuscript cancers-1377410

The authors submitted a guideline that aims to help oncology professionals address the psychosocial aspects of their adult patients and of those who care for them. The idea of ​​the design is good, and an implementation would be valuable.

Some additions and corrections are required to improve the guideline.

Clear objectives must be recognizable for the reader of the guideline. These should be clearly shown in a figure.

It would also be important to point out that psychotherapy is ideally offered in a multimodal therapy setting.

What are the implications. these must be worked out at the end of the discussion.

The limitations of the guideline must be also worked out. Most meta-analysis deal with psychotherapy as a single therapeutic measure, not as part of a multimodal therapy program for cancer patients.

Author Response

We sincerely thank the reviewer for his/her time and for the comments and suggestions which have enabled us to improve our paper. Please see the attachment for a point-by-point response. 
